# Hypomagnesemia as a Risk Factor and Accelerator for Vascular Aging in Diabetes Mellitus and Chronic Kidney Disease

**DOI:** 10.3390/metabo13020306

**Published:** 2023-02-19

**Authors:** Ákos Géza Pethő, Mihály Tapolyai, Maria Browne, Tibor Fülöp

**Affiliations:** 1Department of Internal Medicine and Oncology, Faculty of Medicine, Semmelweis University, 1085 Budapest, Hungary; 2Department of Nephrology, Szent Margit Kórhaz, 1032 Budapest, Hungary; 3Medicine Service, Ralph H. Jonson VA Medical Center, Charleston, SC 29401, USA; 4Department of Medicine, Division of Nephrology, University of Maryland Medical Center, Baltimore, MD 21201, USA; 5Department of Medicine, Division of Nephrology, Medical University of South Carolina, Charleston, SC 29425, USA

**Keywords:** bone-mineral disorders, cardiovascular mortality, magnesium, nutrition supplement, prevention, vascular aging, vascular calcification, vascular smooth muscle cells

## Abstract

The age-old axiom that one is as old as his or her vessels are, calls for ongoing critical re-examination of modifiable risk factors of accelerated vascular ageing in chronic kidney diseases. Attempts to modulate vascular risk with cholesterol-lowering agents have largely failed in advanced chronic kidney disease (CKD). In addition to nitrogen waste products, many pathological biochemical processes also play a role in vascular calcification in chronic kidney damage. Magnesium, a cation vital for the body, may substantially reduce cardiovascular diseases’ risk and progression. This narrative review aimed to address the relationship between hypomagnesemia and vascular calcification, which promotes further cardiovascular complications in diabetes, aging, and CKD. Articles with predefined keywords were searched for in the PubMed and Google Scholar databases with specific inclusion and exclusion criteria. We hypothesized that a decrease in serum magnesium levels contributes to increased vascular calcification and thereby increases cardiovascular mortality. In summary, based on existing evidence in the literature, it appears that simple and inexpensive oral magnesium supplementation may reduce the cardiovascular mortality of patients who are already severely affected by such diseases; in this context, the concept of ‘normal’ vs. ‘ideal’ serum magnesium levels should be carefully re-examined.

## 1. Introduction

Cardiovascular diseases (CVD), such as coronary artery disease, congestive heart failure, arrhythmias, and sudden cardiac death, are leading causes of morbidity and mortality in patients with chronic kidney disease (CKD). In contrast, major cardiac events are presumed to represent almost 50% of the causes of death in CKD patients [1,2]. Kidney disease significantly affects global health as a direct cause of global morbidity and mortality and as a significant risk factor for cardiovascular disease. A meta-analysis of observational studies estimating the prevalence of CKD suggests that approximately 13.4% of the world’s population has CKD [3]. The leading causes of CKD vary by setting, with hypertension and diabetes being the most common causes [4,5,6]. Therefore, treating diabetes and hypertension and reaching the target results can improve renal and cardiovascular outcomes and slow or prevent progression to ESKD (End Stage Kidney Disease) [7,8]. In particular, reasonable glycemic control and anti-hypertensive and hypolipidemic therapy are the cornerstones of the effective treatment of CKD and CVD [9]. In chronic kidney disease, the retention of uremic toxins plays a role in the development of cardiovascular disease and death [10]. Therefore, preserving kidney function can improve outcomes and reduce CVD and CVD-related mortality. This can be achieved through non-drug strategies (e.g., dietary and lifestyle changes) or pharmacological interventions for chronic kidney disease [11]. Based on international guidelines, this paper adopted chronic kidney disease staging based on the estimated glomerular filtration rate (eGFR); we used Kidney Disease Outcomes Quality Initiative (K/DOQI) CKD staging (Table 1) [12]. However, progressive CKD confers escalating healthcare costs commensurate with cardiovascular morbidity and mortality [13]. It indicates the importance of CKD’s prevention and treatment and estimates that the annual global number of deaths caused directly by CKD is between 5–10 million [14]. Currently, the number of people receiving renal replacement therapy exceeds 2.5 million worldwide and is projected to increase up to 5.4 million by 2030 [15]. While most of the mortality burden is mediated by age, dialysis duration, and diabetes mellitus, modifiable risk factors for cardiovascular mortality in CKD patients remain of interest for medical practitioners [16].

Based on the above, in our narrative review, we wanted to explore the significance of hypomagnesemia as a modifiable risk factor. According to our hypothesis, hypomagnesemia plays a role in the development and progression of vascular calcification and should be viewed as a risk factor akin to diabetes, chronic kidney disease, and aging, as well as a contributor to these processes. 

## 2. Search Methods

We searched the PubMed database and Google Scholar using the keywords “hypomagnesemia”, “aging”, “diabetes”, “cardiovascular disease”, “cardiovascular”, “chronic kidney disease”, and “vascular calcification”. We followed the guidelines for literature search and manuscript preparation [17,18]. The inclusion criteria comprised all original articles (human and animal studies) and systematic reviews until 17 November 2022. We did not specify the start time in the search; we only used the first publication available for the given keyword as a basis. We excluded repeated publications. Pertinent publications were located by looking through the references of retrieved articles (backward search) and more recent articles that cited the recovered paper (ahead search).

Studies addressing the objectives of the review and our hypotheses were extracted. Data collection for each study focused primarily on possible connections to our theory. Three authors independently screened each retrieved article for eligibility and extracted study data. Any discrepancies between the first and senior authors (Á.G.P., M.T., and T.F.) were resolved by discussion and agreement.

## 3. Results 

Based on the above, in our narrative review statement, we wanted to draw attention to reducing a risk factor that can be relatively easily addressed, such as hypomagnesemia. The most studied connection was hypomagnesemia and cardiovascular disease.

The announcements corresponding to the search terms are listed in Table 2. It can be seen from the listed publications that the association between hypomagnesemia and other diseases has been a concern of researchers since 1963. Due to the fact that the principal mechanism of cardiovascular diseases is vascular calcification, our interest was mainly focused on the relationship between hypomagnesemia and vascular calcification (Table 3).

From the listed relevant references (Table 3), only a few were animal and experimental investigations (e.g., Chrysant, S.G., et al. [19] and Yao, Z., et al. [20]). Most cited publications summarize the results of human observational studies and case reports. The role of hypomagnesemia in G3-G4 CKD and the chronic hemodialysis patient population was investigated in human studies. Adult male and female patients were included in the observational studies. The role of hypomagnesemia in vascular calcification has not been investigated in children. Several publications summarized the correlations between hypomagnesemia and vascular calcification in a review (e.g., Wei, M., et al. [21], Spiegel, D.M. [22], M de Francisco, A.L., et al. [23], Floege, J. [24], Heaf, J.G. [25], Rosa-Diez, G., et al. [26], Apetrii, M., et al. [27], Van Laecke, S. [28], Rodelo-Haad, C., et al. [29] and Sakaguchi, Y. [30]).

## 4. Cardiovascular Mortality in Chronic Kidney Disease

The increased risk of cardiovascular mortality parallels the deterioration of kidney function, and that cardiovascular risk is particularly evident in CKD patients with stages G 3b–4 (according to the K/DOQI CKD classification) and in those undergoing renal replacement therapy—RRT (hemodialysis, peritoneal dialysis, and transplant) [31,32]. Even mild to moderate kidney damage is associated with significantly higher mortality compared to the healthy population. Parallel to the deterioration of kidney function, many pathological biochemical processes emerge, which contribute to the development of cardiovascular diseases. Such processes include chronic inflammation, insulin resistance, hyperhomocysteinemia, lipid dysmetabolism, and the gradual accumulation of several toxins (aliphatic amines, furans, guanidines, indoles, β2 microglobulin, nucleosides, leptin, parathyroid hormone, phenols, polyols, and endogenous inhibitors of erythropoiesis) [33,34]. In addition to the various toxicants that accumulate, there are specific metabolites and biomarkers that contribute to cardiovascular disease associated with chronic kidney disease, such as asymmetric dimethylarginine, C-reactive protein, homocysteine, ischemia-modified albumin, natriuretic peptides, serum amyloid A, troponin, and type I fibrinolytic zymogen activator inhibitors [35]. Furthermore, despite regular dialysis treatment with chronic hemodialysis or peritoneal dialysis, the expected incomplete removal of organic waste compounds results in the accumulation of uremic toxins, which play a crucial role in the progression of CKD and CVD [36,37]. The possible markers for CVD in CKD are intensively investigated and can also predict coronary artery disease. The levels of matrix Gla protein, neutrophil-lymphocyte ratio, and interleukin 6 (IL-6) were found to correlate with CVD in CKD. It seems that NLR and IL-6 are associated with the increased risk for CVD, and higher matrix Gla protein levels represent a protective factor [38].

Additionally, the gut microbiome is a major source of uremic toxins, including advanced glycation end products, indoxyl-sulfate, and inorganic waste (e.g., phosphate). Those toxins promote vascular calcification by several mechanisms. The most important of these are trans-differentiation and apoptosis of vascular smooth muscle cells (VSMC), dysfunction of endothelial cells, oxidative stress, and interaction with the local renin-angiotensin-aldosterone system or microRNA (micro ribonucleotide acid) profile modification [39]. The therapeutic “gastrointestinal decontamination” is a potential anti-vascular calcification strategy, with the removal of toxins in situ or impediment of absorption within the gastrointestinal tract [40]. First, modulation of the gut microbiota can be achieved by optimizing dietary composition and using prebiotics or probiotics [41]. Other promising strategies include reducing calcium load, minimizing intestinal phosphate absorption by optimizing phosphate binders and inhibiting luminal phosphate transporters, administering magnesium, and using oral toxin adsorbents to remove uremic toxins [42,43]. 

We will briefly summarize that numerous pathological settings are associated with the development of vascular calcification in CKD. The most important contributors are chronic inflammation, oxidative stress, endothelial dysfunction, VSMC trans-differentiation, proliferation, and apoptosis; increased remodeling of the extracellular matrix; loss of mineralization inhibitors; and the release of calcifying extracellular vesicles [44,45]. Until now, the effects of many targeted therapies have been investigated to reduce vascular calcification. However, currently we need more or contradictory data on interventions evaluated in clinical trials to alleviate vascular calcification in patients with CKD [46].

Notwithstanding all these risk factors created in the context of CKD, hypomagnesemia is yet another interesting and underappreciated risk factor for cardiovascular mortality. Observational data have shown an association between low serum magnesium concentrations or magnesium intake and the increased risk of atherosclerosis, coronary artery disease, arrhythmias, aortic valve calcification [47,48,49,50], and heart failure [51]. Furthermore, low serum magnesium levels were associated with an increased risk of end-stage renal disease associated with high serum phosphate levels in a cohort study, suggesting a strong relationship between magnesium deficiency and the adverse effects of high phosphate levels. More importantly, magnesium can strongly inhibit phosphate-induced calcification of VSMCs [52]. A meta-analysis found that oral magnesium supplementation alone in hemodialysis (HD) patients improved chronic kidney disease-mineral-bone disease (CKD-MBD) by modulating serum Ca and parathyroid hormone (PTH) metabolism and improving carotid intima-media thickness reduction [53]. Furthermore, another meta-analysis demonstrated that magnesium concentration is inversely associated with all-cause mortality and cardiovascular mortality and events and magnesium supplementation may improve risk in patients with CKD [54].

However, major supplementation trials with magnesium have reported inconsistent benefits and raised the potential for adverse effects from magnesium overload [55]. Increased urinary magnesium excretion leads to the development of CVD in CKD G-5 [56]. Behind the increased magnesium excretion is the elevated magnesium wasting that generates this seemingly paradoxical risk profile.

## 5. Magnesium Homeostasis 

Magnesium is the most abundant intracellular divalent cation. Magnesium homeostasis has been discussed in detail in numerous publications, so we will focus on only the most important ones. It is essential for maintaining normal cellular physiology and metabolism, acting as a cofactor of multiple enzymes, regulating ion channels and energy generation [57]. Magnesium also regulates vascular tone, atherogenesis and thrombosis, vascular calcification, and the proliferation and migration of endothelial and vascular smooth muscle cells. As such, magnesium can potentially significantly influence the pathogenesis of cardiovascular disease [58]. For example, various conditions could lead to hypomagnesemia (Table 4). In addition, magnesium homeostasis is primarily a factor between intestinal absorption and the kidneys’ regulation of magnesium excretion. Thus, kidney disorders can potentially lead to both magnesium depletion and overload and, as such, increase the risk of cardiovascular disease [59]. 

Magnesium is the fourth most common cation and an essential element for the human body. It plays a vital role in signal transduction, oxidative phosphorylation, glycolysis, protein and deoxyribonucleic acid (DNA) synthesis, and many other biological processes. The vast majority (98%) of total body magnesium (25 g or 1000 mmol) is found in soft tissue (38%) and bone (60%) [60]. Only a small fraction of this is replaceable and can relieve the magnesium deficiency that occurs; the ionized 55–70% serum fraction is biologically active. Under physiological conditions, only 1–2% of the total body magnesium is present in the extracellular fluid [57]. Low magnesium intake is linked to an increased mortality risk and a higher incidence of cardiovascular diseases, diabetes, strokes, cancers, and fractures [61]. Figure 1 illustrates the distribution, absorption, and excretion of magnesium in the body. Additionally, there is a minor degree of active transcellular transport across the channels that transport magnesium, the transient receptor potential cation channel, subfamily M, member 6 (TRPM6), and TRPM7 in the colon [62]. Active vitamin D (calcitriol) increases the absorption of nutrients in the intestines. Much of the magnesium filtered in the kidney is subsequently reabsorbed in the thin limb of Henle’s loop across tight junction channels.

In contrast, the absorption of the remaining filtered magnesium occurs at the proximal tubule level and in the distal collecting tubule, the latter of which is the site of active absorption via TRMP6 [57]. Lower magnesium intake can increase intestinal absorption from 40% to 80%, while the fractional excretion of magnesium in the urine decreases to 0.5%. Hypomagnesemia in chronic kidney failure patients has become prevalent and may be severe because of the increased use of diuretics [63,64], most of which have magnesium-wasting solid side effects. Even in patients on dialysis, magnesium levels [65,66] may be lowered below the desirable limit by the treatment itself. The degree and type of clinical symptoms associated with low or high magnesium levels depend on the serum magnesium concentration, the time of onset, and whether or not there is a concomitant electrolyte disturbance such as calcium or potassium (Table 5). 

## 6. Hypomagnesemia and Cardiovascular Mortality

It is well known that the genetic mutation at 17q12 is related to renal magnesium wasting with co-morbid extensive coronary and vascular calcifications [67]. The first hypomagnesemia-induced vascular calcification was described in an animal model in 1988 by S. G. Chrysant et al. They concluded that dietary-induced hypomagnesemia aggravated hypertension and caused widespread tissue calcification in rats through calcium-mediated systemic vasoconstriction and increased arterial pressure [19]. Several years later, Mingxin Wei et al. documented in the literature review the existence of a significant negative correlation between serum magnesium and intact parathyroid hormone, as well as a negative correlation between serum magnesium and vascular calcification in ESKD patients [21]. This finding is true for hemodialysis (HD) and peritoneal dialysis (PD) patients. They also foresee that vascular calcification could be reduced with an inexpensive magnesium-containing oral phosphate binder. Ishimura et al. demonstrated in a cohort that hypomagnesemia is significantly associated with the vascular calcification of the hand arteries; this association remained independent of calcium and phosphate levels. These authors also theorized that higher serum magnesium levels might have a crucial role in preventing vascular calcification in patients on dialysis [68]. David M. Spiegel also discussed the role of magnesium in chronic kidney failure. He believed that magnesium administered externally may serve as a phosphate binder and may have cardioprotective effects that are associated with vascular and cardiac calcification. To demonstrate these hypotheses, randomized clinical trials are necessary [22]. Angel L. M. de Francisco et al. concluded that moderate hypermagnesemia seems beneficial for vascular calcification and mortality rates in CKD patients. At the same time, magnesium deficiency increases the risk for several diseases, such as type 2 diabetes mellitus, hypertension, and atherosclerosis [23]. 

Andreas Tomaschitz et al. suggested that inappropriately high aldosterone and PTH secretion is strongly linked with CVD development and progression. While hyperaldosteronism increases calcium and magnesium in the urine, a trend of hypocalcemia and hypomagnesemia is present, and secondary hyperparathyroidism causes myocardial fibrosis and disturbs bone metabolism. A blockade of the mineralocorticoid receptor and the removal of the adrenals disrupt this negative feedback loop. This may explain why, following parathyroidectomy, lower aldosterone levels are observed alongside improved patient outcomes [69]. Patrícia João Matias et al. conducted a 48-month-long prospective study of hemodialysis patients using 1.0 mmol/l for dialysate magnesium concentration. The results showed that lower magnesium levels seem to be associated with increased cardiovascular risk markers, e.g., increased pulse pressure, left ventricular mass index, and vascular calcifications, which are linked with a higher mortality rate in hemodialysis patients [70]. The result is thought-provoking: vascular calcification can be reduced by simply increasing the magnesium concentration of the hemodialysis fluid. This led to Jürgen Floege raising the exciting question of whether magnesium would be a calcification inhibitor in chronic kidney failure.

Similarly, both in cell culture studies and clinical situations, low magnesium levels were associated with vascular calcification, cardiovascular disease, and a changed bone-mineral metabolism [24]. This correlation can also be observed in patients with PD. James Goya Heaf reported that elderly PD patients are especially at risk for hypomagnesemia, particularly after long-term PD, due to the combination of decreased dietary intake, protein-energy wasting, and low magnesium levels (0.25 meq/L) in the dialysate solutions, resulting in a negative magnesium balance [25]. Low serum magnesium levels can develop not only for the reasons listed above but also due to drug interactions, such as prolonged use of proton pump inhibitors (PPI). This is a significant concern because of the ubiquitous use of PPI in the current era [71,72,73,74]. However, the severity of PPI-induced hypomagnesemia is also influenced by the dose of PPI [75]. Radojica V. Stolic et al. demonstrated in a cohort that arteriovenous fistula complications significantly correlate with low serum magnesium levels [76]. The quality of arteriovenous fistulas plays a role in the quality of life and the outcome of HD patients. Guillermo Rosa-Diez et al. found that sevelamer carbonate significantly increased serum magnesium levels in hemodialysis patients. This effect persisted even after adjustment for PPI use in the multivariable model [26]. The mechanism by which sevelamer use would improve serum magnesium levels remains unclear. 

Anique D. Ter Braake et al. studied how magnesium can influence vascular calcification. Today, two primary hypotheses exist: the first is that magnesium may bind phosphate and delay the growth of calcium phosphate crystals in the bloodstream; this would allow magnesium to have a passive effect on the deposition of calcium phosphate in the vessel’s walls. Second, magnesium may inhibit vascular smooth muscle transdifferentiation of VSMCs into an osteogenic phenotype by activating a cellular mechanism that promotes calcification [77]. Various theories and observations have led to the conclusion that in ESKD, magnesium supplementation may help lower the serum phosphate level, reduce PTH levels, and interfere with vascular calcification and bone mineralization. It may also reduce the total number of deaths from all causes and cardiovascular diseases [27]. This assumption is confirmed by Zhihui Yao et al.’s animal experiments, where they were able to prevent vascular calcification by using magnesium citrate. Obviously, the cost of *per os* magnesium supplementation would be trivial, when comparing with the enormous expense of kidney replacement therapy with maintenance dialysis.

Clinically, magnesium has been used intravenously in cardiology and obstetric indications before. It is essential to recognize that the most severe hypermagnesemia (Table 3) can be fatal but is primarily observed in patients that consume much magnesium through supplements or magnesium-containing cathartics and antacids, alone or in combination with a major reduction of kidney function [28]. Severe hypermagnesemia can have a neurological presentation, including confusion, absent deep tendon reflexes, articulation issues, gait disturbances, nausea and vomiting, bradycardia, or malignant ventricular tachycardia [78]. However, Sakaguchi et al. documented that those patients on hemodialysis with moderate hypermagnesemia had a survival advantage [30]. Their primary finding was that a lower serum magnesium level was an independent and significant predictor of CVD mortality in chronic hemodialysis patients. 

Additionally, there was a significant correlation between hypomagnesemia and non-CVD mortality, specifically deaths caused by severe infection. Despite hypermagnesemia, total ionized magnesium concentration measured within the normal range. According to this, moderate hypermagnesemia in HD patients ensures normal magnesium levels. Dieter Haffner et al. also highlighted that it is necessary to emphasize sufficient magnesium intake not only in renal replacement therapy but also after kidney transplantation [79]. Post-transplant CKD-MBD is primarily caused by preexisting renal osteodystrophy, cardiovascular issues during the transplant process, and the use of glucocorticoids. Rodelo-Haad et al. described the potential role of magnesium in cardiovascular calcification. It seems that magnesium may reduce vascular calcification by the direct inhibition of the Wnt/β-catenin signaling pathway [29,80]. In vascular calcification, calcium and phosphate deposition play a crucial role, leading to these multiple pathological processes. The chronic inflammation and increased expression of pro-inflammatory cytokines (e.g., Tumor necrosis factor-alpha [TNF-α], interleukin [IL]-1β, IL-6, cyclooxygenase [COX]-2, and inducible nitric oxide synthetase [iNOS]) initiate the osteogenic transdifferentiation of VSMCs into calcifying cells in the vasculature [81]. Bone morphogenetic proteins BMP signaling regulators such as MGP and BMP-binding endothelial regulator proteins serve a protective role in vascular calcification [82]. Vitamin K-dependent MGP is a crucial inhibitor of vascular calcification (VC), which is mediated by magnesium (Figure 2). It appears that neutrophil-to-lymphocyte ratio and IL-6 are associated with increased risk for CVD, and higher MGP levels represent a protective factor [38]. From this consideration, we can also understand why oral magnesium supplementation could be an effective “anti-vascular calcification” drug.

Furthermore, magnesium deficiency accelerates the atherosclerotic process. Ram et al. demonstrated that after heart transplantation, drug-induced hypomagnesemia is associated with cardiovascular calcifications but not the use or nonuse of statins [83]. This limitation is crucial because currently, statins are the default agents used to prevent and treat coronary heart disease. 

Additionally, magnesium deficiency exacerbates the detrimental effects of an increased tubular phosphate load if kidney injury is already present. An adverse effect of excess tubular phosphate loading is reduced expression of α-Klotho in renal tubules of patients with mild to moderate CKD. Low magnesium levels exacerbate the Klotho depletion caused by the phosphate’s tubular load [29]. Ionized magnesium is a biologically and physiologically active form, and the concentration of serum ionized magnesium is difficult to assess in clinical practice. Christopher Holzmann-Littig et al. developed and internally validated an equation to estimate ionized magnesium from routinely assessed serum variables and demographic data. Their simple equation contains only three variables: (−0.129 + (0.620 × Mgtot) + (0.337 × Caion) + (−0.110 × Catot) (Mgtot: total magnesium, Catot total calcium, and Caion: ionized calcium)) [84]. Knowing the exact serum magnesium level cannot be neglected because this might be a way to reduce the risk of cardiovascular death [85]. The cause of the hypomagnesemia and the correlation with CVD in PD are unclear. Since lower magnesium serum levels are associated with poor nutritional status and increased inflammation, the association between serum magnesium and mortality in patients with PD should be further investigated before a definite conclusion is reached. A treatment strategy that maintains the appropriate serum magnesium level should be devised [86].

All the above discussion and listing of the association between hypomagnesemia and vascular calcification focused on the direct effects of VSMCs on calcifying vascular cells. However, there is an additional link with dyslipidemia, as hypomagnesemia worsens the serum lipid profile, further potentiating vascular calcification [87]. 

## 7. Hypomagnesemia and Other Aspects of Health

Numerous clinical studies and empirical evidence have drawn attention to the vital role played by magnesium (Figure 3). Magnesium is crucial for normal insulin sensitivity. W. H. Kao et al. demonstrated that low serum magnesium levels strongly correlate with developing type 2 diabetes mellitus, both in animal studies and human observation [88]. Furthermore, serum magnesium levels negatively correlate with fasting blood glucose and postprandial blood sugar values [89]. Based on this consideration, oral magnesium can improve insulin resistance and stabilize blood glucose levels [90]. Orchard suggested that, in addition to the traditional dietary approach, magnesium supplementation should also be included in diabetes management strategies [91]. Both high blood glucose and high insulin levels are known to increase the excretion of magnesium in the urine. In contrast, the excretion of magnesium and the fasting blood glucose are oppositely related to the serum magnesium level [92]. Normal magnesium homeostasis is integral to the regulation of insulin signaling, insulin receptor kinase phosphorylation, insulin post-receptor actions, insulin-mediated cellular glucose uptake, and glucagon regulation [93]. Mario Barbagallo et al. showed that in type 2 diabetes patients, daily magnesium administration restored a more appropriate intracellular magnesium concentration and contributed to improving insulin-mediated glucose uptake. This suggests that a high daily magnesium intake may lower the incidence of type 2 diabetes [94]. Specific studies of elderly patients have documented the adverse effect of hypomagnesemia on bone health, adequate glycosylation and metabolic compensation, correct cardiac and vascular functioning, and a possible psycho-cognitive profile. As a result, preventing hypomagnesemia may positively impact the aging process of elderly patients [95]. A meta-analysis showed that oral magnesium supplementation and appropriate dietary patterns improve insulin sensitivity and metabolic control in individuals with type 2 diabetes [96].

More recent findings regarding the direct association between magnesium status and immune function suggest magnesium may play another critical role in immune defense mechanisms. In many of the patients who were hospitalized with the SARS-CoV-1 infection, hypomagnesemia was present on admission and tended to worsen during their hospitalization. A magnesium deficiency may augment the inflammatory response, which contributes to the cytokine storm involved in developing severe SARS-CoV-1 symptoms [97]. A large concentration of magnesium bound to lymphocytes (greater than 10 mM) does not “buffer” the external decrease. This results in reduced activation of T-cells and a diminished immune response, as demonstrated in an animal experiment [98]. Epidemiologic studies have shown that a diet low in magnesium is associated with an increased risk of cancer, underscoring magnesium’s significance in hematology and oncology. Hypomagnesemia at diagnosis is associated with a worse prognosis in solid malignant tumors [99]. This statement is true for hematologic malignancies, too. 

Animals and humans are known to rapidly develop respiratory failure and die within a few days when exposed to 100% oxygen. Another interesting fact is that high-dose magnesium therapy tends to attenuate the adverse effects of hyperoxia [100]. Hypomagnesemia is common but is frequently disregarded in critically ill patients in the ICU. However, hypomagnesemia appears to be linked to a higher mortality risk, sepsis, mechanical ventilation, and a longer length of stay in the ICU for patients admitted to the ICU [101]. Osteoporosis is also correlated with hypomagnesemia [102]. 

Hypomagnesemia is a frequent consequence of the refeeding syndrome. Refeeding syndrome is a dangerous condition that can even lead to death. The syndrome occurs following the return of proper nutrition to malnourished and cachectic patients [103]. The leading symptom is hypophosphatemia, often accompanied by electrolyte disturbances, including hypomagnesemia and hypokalemia, along with deficiencies of vitamins and trace elements. Due to the concomitant administration of carbohydrates and intravenous fluid volume, this may also lead to hypervolemia with cardiac failure [104,105]. Refeeding syndrome can be effectively managed and prevented if its predisposing factors and pathophysiology are understood. The initial assessment of thiamine levels and serum electrolytes, including phosphate and magnesium, as well as their supplementation if necessary and a gradual increase in nutritional intake, are all factors crucial to the patient’s well-being [105].

## 8. Conclusions

Based on the cited reports, a reduced serum magnesium level is closely related to the risk of developing vascular calcification, type 2 diabetes, and increased cardiovascular mortality. Current medical therapies utilized in the context of CKD are increasing the likelihood of hypomagnesemia, including in those in need of renal replacement therapy. All these results question the validity of ‘normal’ vs. ‘ideal’ serum magnesium levels to mitigate calcification and premature vascular aging. Our review specifically focused on hypomagnesemia and vascular calcification as the underlying factors of cardiovascular diseases. We can clearly state that vascular calcification is the starting point for both diabetes complications and cardiovascular diseases, including the cardiovascular complications of chronic kidney disease and aging. While it is evident that hypomagnesemia is deleterious, the question remains if the replacement of magnesium can mitigate the undesirable hard endpoints. Prospective studies are recommended to verify the conclusions that can be drawn from the results so far and to verify the risk reduction achieved with oral magnesium supplementation. By applying all this knowledge, the easily measured magnesium concentration in serum is essential in making further therapeutic decisions.

## Figures and Tables

**Figure 1 metabolites-13-00306-f001:**
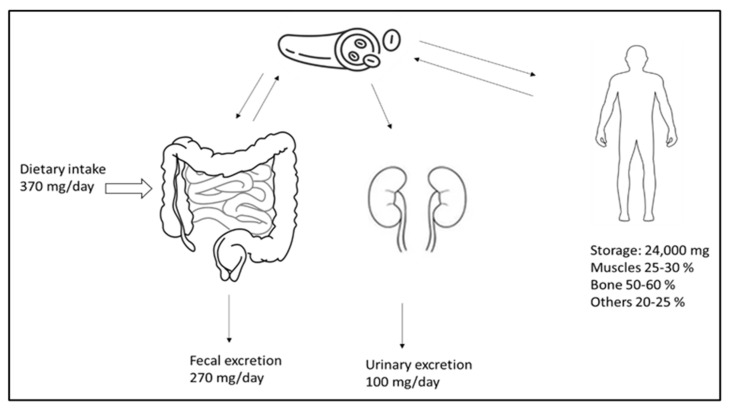
Magnesium homeostasis. The magnesium intake, excretion, and distribution in the body. Adequate intake covers the daily loss of magnesium.

**Figure 2 metabolites-13-00306-f002:**
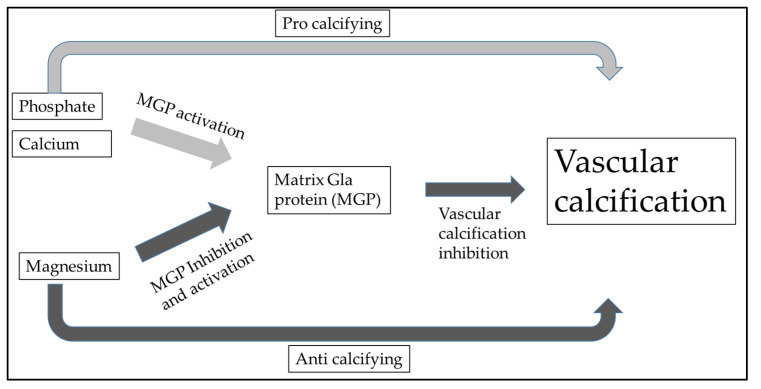
Vitamin K-dependent matrix Gla protein (MGP) is a crucial inhibitor of vascular calcification (VC) and is regulated with magnesium. This also shows that the appropriate magnesium level contributes to the prevention of vascular calcification.

**Figure 3 metabolites-13-00306-f003:**
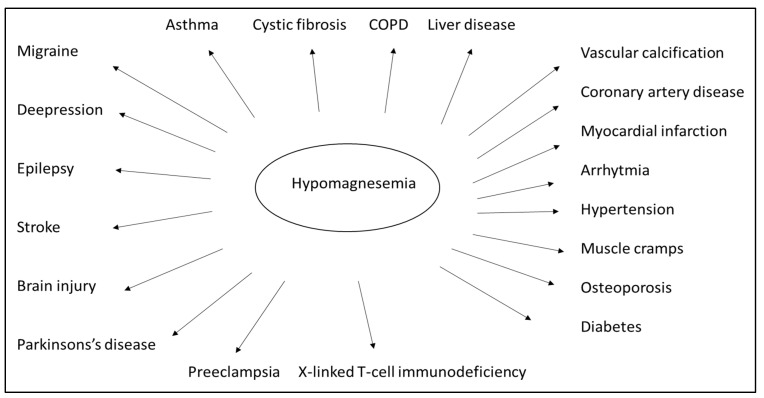
Low levels of magnesium in the body can be associated with many diseases. However, many diseases can be prevented and treated by preventing and correcting hypomagnesemia. (Chronic obstructive pulmonary disease–COPD).

**Table 1 metabolites-13-00306-t001:** The K/DOQI-CKD staging depends on the GFR (glomerular filtration rate). The stadiums between Grades 1–5 (G1–5) depend on the eGFR.

GFR Category	Glomerular Filtration Rate (GFR) mL/min/1.73 m^2^	Description
G1	≥90	Normal
G2	60–89	Mildly decreased
G3a	45–59	Mildly to moderately decreased
G3b	30–44	Moderately to severely decreased
G4	15–29	Severely decreased
G5	<15	Kidney failure

**Table 2 metabolites-13-00306-t002:** The search results for the keywords “hypomagnesemia”, “aging”, “diabetes”, cardiovascular disease”, “cardiovascular”, “chronic kidney disease”, and “vascular calcification”.

Searching Keywords	Number of Publications until 17 November 2022	First Publication in This Field
hypomagnesemia and aging	44	1980
hypomagnesemia and diabetes	593	1963
hypomagnesemia and cardiovascular disease	809	1969
hypomagnesemia and cardiovascular	469	1963
hypomagnesemia and chronic kidney disease	266	1973
hypomagnesemia and vascular calcification	26	1988

**Table 3 metabolites-13-00306-t003:** The relevant references with the association between hypomagnesemia and vascular calcification.

	Authors	Publication Years	Conclusions	Population	Type of the Study
1.	Chrysant, S.G., et al.	1988	Hypomagnesemia aggravated hypertension, widespread tissue calcification, and increased peripheral vascular resistance.	animal	experimental
2.	Honavar, S.G., et al.	2001	Sclerochoroidal calcification is associated with hypomagnesemia.	human	case series
3.	Wei, M., et al.	2006	There is an inverse relationship between serum Mg and vascular calcification.	human	literature review
4.	Ishimura, E., et al.	2007	Hypomagnesemia is significantly associated with the presence of vascular calcification.	human	cohort
5.	Di Iorio, B.R., et al.	2009	Randomized, multicenter, prospective, interventional study. Conclusion: sevelamer improves the survival	human	randomized trial
6.	Spiegel, D.M.	2011	Exogenous administration of the magnesium may be helpful as a phosphate binder.	human	review
7.	M de Francisco, A.L., et al.	2013	It is time for a re-examination of the role of magnesium in CKD patients.	human	review
8.	Tomaschitz, A., et al.	2014	Due to PTH elevation, the magnesiuretic effects are also increased.	human	review
9.	João Matias, P., et al.	2014	Lower Mg levels seem to be associated with higher mortality in HD patients.	human	cohort
10.	Floege, J.	2015	Magnesium has also been linked to diseases such as metabolic syndrome, diabetes, hypertension, fatigue and Depression, all of which are common in CKD	human	review
11.	Misra, P.S., et al.	2015	Among HD patients, proton pump inhibitors users have lower serum Mg levels as compared with non-users	human	cross-section study
12.	Heaf, J.G.	2015	Hypomagnesemia is common in PD and can be treated with magnesium supplements.	human	review
13.	Stolic, R.V., et al.	2016	Hypomagnesemia is a significant pro-atherogenic factor.	human	cohort
14.	Rosa-Diez, G., et al.	2016	Hemodialysis patients receiving sevelamer show higher serum magnesium levels and a reduced risk of vascular calcifications.	human	cohort, review
15.	Ter Braake, A.D., et al.	2017	Magnesium may regulate vascular smooth muscle cell trans-differentiation toward an osteogenic phenotype.	human	review
16.	Apetrii, M., et al.	2018	Normal serum magnesium levels may represent a plausible option to improve the outcome of dialysis patients.	human	review
17.	Tangvoraphonkchai, K., et al.	2018	Hypomagnesemia has been proven or suspected as a cause of cardiac arrhythmias.	human	review
18.	Okamoto, T., et al.	2018	Proton pump inhibitors associated with vascular calcification in patients undergoing dialysis through hypomagnesemia	human	cohort
19.	Yao, Z., et al.	2018	Magnesium may be a potential drug for preventing vascular calcifications in patients with chronic renal failure.	animal	experimental
20.	Van Laecke, S.	2019	Magnesium deficiency and/or hypomagnesemia have been linked to cardiovascular disease and vascular calcification.	human	narrative review
21.	Li, H.J., et al.	2019	Hyperparathyroidism and hypomagnesemia may contribute to significant cardiovascular risk.	human	case report
22.	Rodelo-Haad, C., et al.	2020	Low serum magnesium is associated with unfavorable clinical outcomes such as major adverse cardiovascular and renal events.	human	review
23.	Haffner, D., et al.	2021	The role of magnesium supplementation after kidney transplantation. Conclusion: Hypomagnesemia could develop after kidney transplantation	human	cohort
24.	Holzmann-Littig, C., et al.	2021	The importance of magnesium concentration in hemodialysis patients. Conclusion: With an easy equation could prevent HD patients from hypomagnesemia	human	cohort
25.	Liu, H., et al.	2021	Hypomagnesemia is significantly associated with cardiovascular and all-cause mortality in maintenance HD patients.	human	meta-analysis
26.	Sakaguchi, Y.	2022	Magnesium might provide a better cardiovascular prognosis.	human	randomized, cohort, review

Abbreviation: CKD, chronic kidney disease; HD, hemodialysis; Mg, magnesium; PD, peritoneal dialysis; PTH, parathyroid hormone

**Table 4 metabolites-13-00306-t004:** The causes of hypomagnesemia. Hypomagnesemia may result from either reduced intake and bioavailability or increased renal and gastrointestinal loss.

Origin of Loss	Etiology	Possible Diseases
Extrarenal	GI-loses	IBD, bariatric surgery, vomiting, malignancy of GI tract
	Decreased GI absorption	IBD, bariatric surgery, previous intestinal resection, decreased intake, drugs, vitamin D deficiency
	Others	Sepsis, blood transfusion, hungry bone syndrome, refeeding
Renal	Hereditary	Hypercalciuric hypomagnesemia, Gitelman-like hypomagnesemia, mitochondrial-related hypomagnesemia
	Drugs	Platinum derived anti-cancer agents, diuretics, aminoglycosides, EGFR inhibitors, calcineurin inhibitors, pentamidine
	Miscellaneous	Acidosis, osmotic diuresis, proteinuria, polyuria, insulin resistance, alcoholism, primary and secondary hyperparathyroidism

Abbreviations: EGFR: epidermal growth factor receptor, GI: gastrointestinal, IBD: inflammatory bowel disease

**Table 5 metabolites-13-00306-t005:** Clinical symptoms of dysmagnesemias. The severity of the clinical signs depends on the serum magnesium concentration (conversion between mmol/L and mg/dL: multiply by 2.43).

Serum Magnesium (mmol/L)	Clinical Symptoms
<0.4	tetany, nystagmus, seizures, psychosis, arrhythmia
0.4–0.7	tremor, neuromuscular irritability, hypokalemia, hypocalcemia
0.7–1.1	normal range
>2.0	lethargy, flushing, nausea, vomiting, diminished deep tendon reflexes, bradycardia, drowsiness
>3.0	tachycardia, slurred speech, muscle weakness, EKG changes
>4.0	hypotension, confusion, loss of deep tendon reflexes, quadriparesis, respiratory depression
>5.0	coma, apnea, cardiac arrest, asystole, death

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
