# Peer review of "Hypomagnesemia as a Risk Factor and Accelerator for Vascular Aging in Diabetes Mellitus and Chronic Kidney Disease"

_metabolites, 2023, doi:10.3390/metabo13020306_

Round 1
Reviewer 1 Report
Despite being a bibliographical review, the search design, the results, and their discussion are not related, and a conclusion cannot be reached. It is suggested to review the design in detail, present the results found per article evaluated and that the evaluation/analysis of these adhere to the main objective that led them to carry out this bibliographic review. The main objective would be to address "the relationship between hypomagnesemia and vascular calcification" in patients with this CKD-CVD-DM triad".
1. The manuscript needs writing and language editing. The title should be improved, for example, “Hypomagnesemia like a vascular risk factor in this triad of chronic kidney disease-cardiovascular disease-diabetes mellitus”. The abstract should present the main point of this literature review. The main aim must be direct and the same throughout the manuscript (abstract, introduction, results/discussion). Authors should not use the words that appear in the title as keywords. References should be recent and relevant, they should be well referenced, and their use should be improved throughout the manuscript.
2. The intro section should improve. A proper presentation and a good and clear justification (reason) for conducting this review study should be given. Why is the magnesium level important in this "CVD-DM-CKD" triad? It would be better if the authors offered a hypothesis rather than the main objective of this study. What was the main objective of this study? Line 60. The first time an abbreviation appears, the full name must be entered. What does it mean? for example, G1 in Table 1? Line 70: It would be a good idea to write the full name here.
3. The methods section is sparse and needs to be improved. The description must be clear, concise, and detailed. The authors must declare that this review of the literature was carried out in accordance with the principles of the Declaration of Helsinki on data management. The authors must declare that this review of the literature was carried out in accordance with the principles of the Declaration of Helsinki on data management. How was the review carried out (give details)? What were the inclusion and exclusion criteria? In which group of people was this review conducted? What information was collected: type of population, type of study, year of publication, age range, type of sample (serum?), the cut-off points for normal levels of magnesium, etc. (give details)? All variables studied should be described, defined, and measured appropriately. Line 77: What year did the search start?
4. It would be a good idea to add a results section: How many articles did this search return? What were the most interesting and significant results? How was information about the association/relationship between hypomagnesemia and vascular calcification sought and collected? It would be better if the authors summarized them in a results table of this search. Can the authors show their results by percentages? It should be clear which were the most significant results collected from this literature review.
5. The discussion section should start with the main objective of this review study and the most significant results found. The collected results (subsections) by the authors should be discussed from multiple angles and placed in context without overinterpreting them. Line 168: In the general population? A paragraph of limitations and suggestions for this study should be written before the conclusion.
6. The conclusion must be the same throughout the manuscript. The introduction, the study design, and the discussion of the results should lead the reader to the same conclusion as the authors.
I would like to encourage the authors to rewrite this narrative review, thinking about the main objective of this study, and its design and responding with the results and arguments of the discussion to the most appropriate conclusion of this research work.

Author Response
Answers enclosed.

Reviewer 2 Report
There seems to be a discordance between the title of the paper “Hypomagnesemia as a risk factor for vascular aging in diabetes mellitus and chronic kidney disease” and partially the content.
The authors discuss about vascular calcifications but yet again the overview of vascular calcifications in diabetes and chronic kidney disease is too little overviewed in the paper.
Figure 1 – there is a discordance between the text and the figure.
Bones are not soft tissue, please correct.
Section 5 – the authors discuss mostly about hypomagnesemia and vascular calcifications, not about cardiovascular mortality as advertised in the title of this section. Magnesium is also linked to dyslipidemia and serum lipids’ concentrations in cardiovascular disease, diabetes and CKD and this should also be briefly discussed. See: https://www.mdpi.com/2072-6643/13/5/1411
Please create a summative table for your paper with at least the following sections: author of the study, year of publication, data on vascular calcifications related to hypomagnesemia in diabetes mellitus and chronic kidney disease, mechanisms via which hypomagnesemia contributed to the development of vascular calcifications in these disorders, conclusions, practical applications.
The general information in the article should be reduced as there is nothing novel about CKD staging, magnesium homeostasis etc and instead you should focus on the topic of the paper – vascular aging/calcifications and their relationship with hypomagnesemia in diabetes and chronic kidney disease.
Author Response
Answers enclosed.

Round 2
Reviewer 1 Report
This narrative review manuscript has improved considerably, but there are some important points that need clarification before publication. It would be nice if the authors describe more details about the 26 articles found since 1963, for example, if they were done on animals, and if they were done on humans: in which age group (adults, elderly, etc.) in both sexes? Was any done in children or adolescents? What kind of studio were they? retrospective, prospective, follow-up, etc.
Table 3, article 5: What were the conclusions of this study? Abbreviations: PD meaning. Line 165: The first time an abbreviation appears, the full name must be entered, such as VSMCs, HD, PTH, TRPM, etc. Line 224 and 226: Table 5. Line 312: It would be better in another paragraph. Lines325-326: (e.g., Tumor necrosis factor-alpha [TNF-α], interleukin [IL]-1β, IL-6, cyclooxygenase [COX]-2, inducible nitric oxide synthase).

Author Response
This narrative review manuscript has improved considerably, but there are some important points that need clarification before publication. It would be nice if the authors describe more details about the 26 articles found since 1963, for example, if they were done on animals, and if they were done on humans: in which age group (adults, elderly, etc.) in both sexes? Was any done in children or adolescents? What kind of studio were they? retrospective, prospective, follow-up, etc.
Answer: Thank you for your effort and suggestions. I added a brief description of the listed relevant references.
Table 3, article 5: What were the conclusions of this study? Abbreviations: PD meaning. Line 165: The first time an abbreviation appears, the full name must be entered, such as VSMCs, HD, PTH, TRPM, etc. Line 224 and 226: Table 5. Line 312: It would be better in another paragraph. Lines325-326: (e.g., Tumor necrosis factor-alpha [TNF-α], interleukin [IL]-1β, IL-6, cyclooxygenase [COX]-2, inducible nitric oxide synthase).
Answer: Thank you for the advised corrections. I made all of the corrections in the manuscript.
Reviewer 2 Report
The revised manuscript is satisfactory, the authors have addressed my concerns.
Author Response
Answer: Thank you for your effort and suggestions.